

# The effects of gluteal stretching *vs.* Lightback® on hip rotation range of motion and posterior chain flexibility in healthy subjects: a cross-over clinical trial

Charles Cotteret, Jaime Almazán-Polo, Fabien Guérineau and
Ángel González de-la-Flor

Department of Physiotherapy, Faculty of Medicine, Health and Sport, Europea University of
Madrid, Villaviciosa de Odón, Madrid, Spain

## ABSTRACT

**Introduction:** Hip range of motion (ROM) across various planes is necessary in sport-related activities. Static stretching was commonly used to improve hip ROM. The Lightback system, a novel compression device, has been developed to enhance hip mobility by applying controlled axial forces on the femur. This study aimed to evaluate the effectiveness of the Lightback system in improving hip ROM and posterior chain flexibility compared to conventional static stretching.

**Methods:** A randomized cross-over trial was carried out in 31 physically active participants (62 lower limbs; $n = 31$ Lightback group (LBG) and $n = 31$ stretching group (SG)). Hip rotation at two positions of hip flexion (active and passive external (ER) and internal rotation (IR) at 0–90° hip flexion), total rotation ROM (TRROM), and the flexibility of the posterior chain (active knee extension test (AKE) and active straight leg raise (ASLR)) were measured before and after the stretching session.

**Results:** LBG demonstrated significantly greater improvements in hip ER and IR compared to the SG. Specifically, the LBG showed significant increases in active ER at 0° ($p = 0.002$) and 90° ($p < 0.001$) of hip flexion, as well as IR at 0° ($p = 0.007$) and 90° ($p < 0.001$). TRROM in neutral and at 90° of hip flexion also improved significantly in the LBG ($p < 0.001$). In passive ROM, the LBG exhibited significant increases in ER at 0° ($p < 0.001$), IR at 90° ($p = 0.001$), and TRROM at both positions ($p < 0.001$), compared to the SG. Regarding posterior chain flexibility, both groups improved in AKE and ASLR ($p < 0.001$), but the LBG showed a significantly larger effect in ASLR ($p < 0.001$), with no significant difference between groups in AKE.

**Conclusion:** This study demonstrated that both the Lightback system and static gluteal stretching improved passive hip ROM. However, the Lightback system showed greater improvements in active ROM, particularly in external and internal rotation at various degrees of hip flexion, as well as in posterior chain flexibility. Notably, the Lightback system significantly enhanced large improvement in the ASLR test.

Corresponding author
Ángel González de-la-Flor, angel.
gonzalez@universidadeuropea.es

## INTRODUCTION

Sports related activities such as running, or pivoting or squatting, implicate hip range of motion (ROM) in different planes. Conventional static stretching was commonly prescribed to address limitations in hip ROM, particularly in several musculoskeletal disorders (*Wicke, Gainey & Figueroa, 2014*; *Bae, Kim & Sung, 2017*; *Tak et al., 2017*; *Gradoz et al., 2018*; *Hatefi, Babakhani & Ashrafizadeh, 2021*). Static stretching, a traditional treatment technique, involves methods like static or ballistic stretching, where active voluntary contraction of the antagonist muscle group exerts tension on the target muscle (*Chaabene et al., 2019*). Despite some controversy regarding the effect of static stretching on force and tension generation, sustained tension has been linked to improved flexibility and the prevention of injuries, particularly in recreational sports (*Chaabene et al., 2019*; *Hatefi, Babakhani & Ashrafizadeh, 2021*). In addition, conventional gluteal stretching techniques often require a substantial ROM in hip rotation to effectively target the deep gluteal muscles. However, individuals with restricted hip rotation ROM may experience challenge to achieve the necessary positions for effective stretching, potentially limiting the benefits of these exercises. This limitation is particularly problematic for athletes or individuals with pre-existing abnormal hip morphology, where achieving the full range of hip rotation may not be feasible (*Audenaert et al., 2012*). As a result, the effectiveness of conventional gluteal stretching in such populations can be compromised, necessitating alternative approaches that can provide similar or superior benefits without requiring large hip rotation ROM (*Bennell, Buchbinder & Hinman, 2015*).

Given the limitations of conventional gluteal stretching, particularly for individuals with restricted hip rotation, alternative approaches that can deliver the same or superior benefits without requiring extensive ROM are necessary. One such innovative solution is the Lightback system, which provides a novel method of enhancing hip mobility and targeting deep gluteal muscles without the need for large hip rotation. It is a mechanical device designed to apply controlled axial forces on the femur in an antero-posterior direction with lateral-medial variations. This device is intended to simulate manual joint therapy techniques by generating posterior glides on the femur, which may help alleviate restrictions in the posterior hip region and improve overall hip mobility. Similarly to manual joint therapy techniques, a posterior glide is generated on the femur, which indirectly stimulates the posterior capsular aspect of the coxofemoral joint. As a self-application system through simple instructions, the patient can develop on his own a force vector like that described in anterior to posterior coxofemoral joint articular techniques, without needing the help of a therapist. Hence, the tension maintained on the deep gluteal muscles through axial forces could be like the stress derived from the static stretching of the posterior structures of the pelvis in hip flexion and external rotation (*Hatefi, Babakhani & Ashrafizadeh, 2021*).

The variability in total rotation ROM (TRROM) between individuals, influenced by anatomical factors such as the femoral head and acetabulum, further underscores the need for tailored interventions (*Short, Macdonald & Strack, 2021*). The TRROM, defined as the sum of the total range of motion in internal and external rotation, has been evaluated in
different positions showing fair to excellent reliability in positions such as supine or sitting (*Gradoz et al., 2018*). Furthermore, the functional differences related to the restriction of total mobility of the hip modify the muscle base length tension of deep muscle groups of the hip, leading to alterations in neuromuscular activation and force production mechanisms (*Nguyen et al., 2017*). Despite the consideration of total hip ROM restriction and posterior chain flexibility as a risk factor for pain (1), there is a lack of evidence that determines and compares the effect of different conservative approaches on the hip ROM and posterior chain.

Thus, the main objective of the study is to compare the effectiveness of the application of Lightback compared to a conventional static stretching of the deep gluteal muscles in the hip external and internal rotation ROM, in hip TRROM, and the posterior chain flexibility through the active knee extension (AKE) test and active straight leg raise (ASLR) test in healthy young adults. We hypothesized that the Lightback Group (LBG) presents greater improvements in hip ROM and flexibility of the posterior chain compared to the static stretching group (SG).

## METHODS

### Study design

A randomized cross-over clinical trial was carried out with a cohort of healthy and physically active participants, adhering to the ethical principles delineated in the Declaration of Helsinki. This study adhered to the established guidelines detailed in the Consolidated Standards of Reporting Trials (CONSORT) (*Dwan et al., 2019*). All the participants signed the informed consent, the study received ethical approval from the European University of Madrid Research Ethics Committee (CIPI: 2023-296), and its protocol was registered with the Australian New Zealand Clinical Trials Registry (ANZCTR: ACTRN12624000300572). Both the 12 years experienced evaluator measuring primary and secondary outcomes and the statistician were blinded to the intervention group allocation.

### Sample size calculation

The determination of the sample size was carried out through the utilization of G*Power Software (v.3.1.9.2). An alpha error rate of 0.05, a statistical power of 0.95, a medium effect size (f = 0.25 or Eta partial squared = 0.06) for the primary outcome measure, the active hip external rotation (ER) at 90° of hip flexion (*Hatefi, Babakhani & Ashrafizadeh, 2021*). The power analysis was performed between-within interaction in a repeated measures with two groups and two times of measurement (baseline and immediate post-intervention). Therefore, at least of 27 participants ($n = 54$ hips) were necessary for the study.

### Participants

The study was conducted with a cohort of physically active and healthy students of the European University of Madrid. Recruitment efforts encompassed the distribution of flyers, posting of posters, and strategic placement of advertisements within the university

premises. Inclusion criteria for participation were both male and female individuals, aged between 18 and 30 and engaging in a consistent training routine of at least 2 days per week. Exclusion criteria included individuals with a recent history, within the last 5 years of musculoskeletal conditions affecting the lower limbs or lumbopelvic region, as well as those with neuromuscular, rheumatic, cardiovascular, or neurological disorders. Additionally, individuals with a history of prior surgical interventions or fractures in the lower extremities or abdominal region were excluded from the study.

## Randomization

Participants were randomly assigned to receive either the LBG or SG on one leg, with the opposite leg receiving the alternative intervention in a 1:1 allocation. Randomization was performed using a computer-generated random sequence (Microsoft Excel). A washout period was not required due to the immediate nature of the outcome measures.

## Procedure

The study included two intervention protocols: the Lightback group (LBG) and the gluteal stretching group (SG). Each intervention was designed and implemented following the Consensus on Exercise Reporting Template (CERT) guidelines (*Slade et al., 2016b*, *2016a*).

### Lightback group

Participants in the LBG received instrumental passive stretching using the Lightback machine (Lightback, Colombelles, France) (Fig. 1). The intervention protocol was as follows:

Participants were seated on the Lightback machine with one leg extended and the other leg positioned in 90 degrees of hip flexion, with the knee in a submaximal flexed position. The pelvic position was continuously monitored to ensure that both ischial tuberosities remained in contact with the ground, maintaining consistent alignment throughout the intervention. A backrest was also utilized to stabilize the torso and ensure that the 90-degree hip flexion position was maintained consistently during the intervention (Fig. 2).

The intervention consisted of six sets of 10-s passive stretches, with 5-s rest intervals between sets. Participants were instructed to focus on maintaining a relaxed state during the stretch to avoid any voluntary muscle contraction that could interfere with the passive stretching effect.

### Gluteal stretching group

Participants in the SG performed a conventional static gluteal stretch. The protocol for this group was as follows:

The gluteal stretch was performed with the participant in a supine position. The leg to be stretched was placed in flexion, abduction, and external rotation, with the ankle and distal tibia resting on the contralateral femur. The stretch was intended to target the deep gluteal muscles, particularly the piriformis and surrounding musculature.

To ensure consistency, a backrest was used to maintain the participant's torso in a stable position, ensuring the hip remained at 90 degrees of flexion during the stretch. The

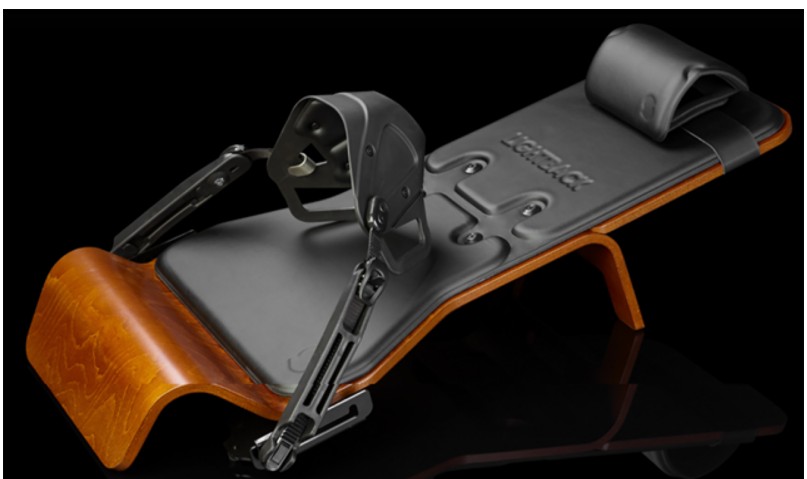

**Figure 1 Lightback instrument.**         

positioning of the pelvis was also monitored to ensure that both ischial tuberosities remained in contact with the ground, preventing any compensatory movements that could alter the effectiveness of the stretch. To ensure patients experienced the proper stretch sensation, they were allowed to adjust external hip rotation while maintaining 90 degrees of hip flexion and abduction (Fig. 3).

The gluteal stretch was performed in two sets, each lasting 30 s, with a 30-s rest period between sets. Participants were instructed to hold the stretch at a point where they felt a mild to moderate stretch, avoiding any discomfort or pain.

Prior to the intervention, participants of the LBG received detailed verbal and visual instructions on how to position themselves correctly on the machine. In addition, participants of the SG received standardized instructions on how to perform the stretch correctly. A trained, 14 years' experience physiotherapist supervised the sessions, providing real-time feedback and adjustments to ensure proper form. The physiotherapist also ensured that the stretch was being performed within a safe range of motion.

Participants' adherence to the protocol was monitored by the supervising physiotherapist, who ensured that the stretch was held consistently for the prescribed duration. Participants were asked to rate their perceived exertion and any discomfort they experienced during the stretch, which was recorded to monitor compliance and adjust the intervention as needed. To standardize the conditions, no warm-up or prior stretching was conducted before data collection. Participants were instructed to carry out their regular daily activities before the intervention sessions. All sessions were conducted at the same time of day to minimize the impact of diurnal variations in flexibility and muscle tone. A washout period was not required due to the immediate nature of the outcome measures.

The primary and secondary outcomes (hip ROM, AKE, and ASLR) were assessed immediately before and after the intervention sessions, with the same procedures applied to both groups.

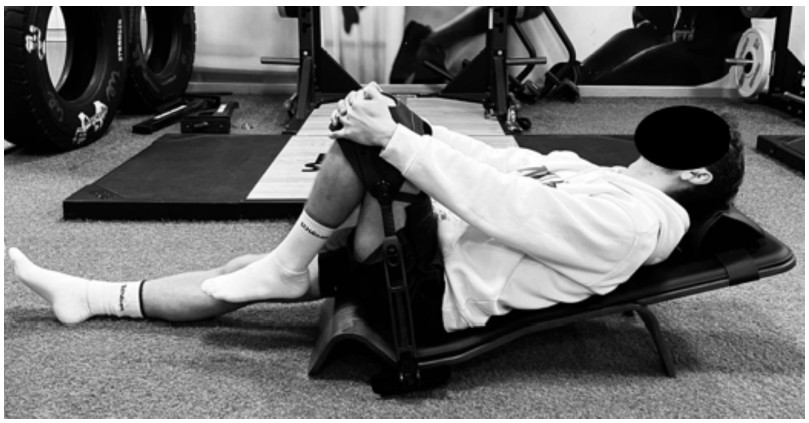

**Figure 2  Lightback stretching group (LBG).**

## Descriptive variables

The sociodemographic data regarding the participant's gender (categorized as male or female), age (expressed in years), height (measured in centimeters), weight (recorded in kilograms), and body mass index (BMI), calculated in kg/cm$^2$ using Quetelet's index (*Garrow, 1986*). Furthermore, information on the dominant limb (identified as right or left) distribution, femur length (*Davis et al., 2006*), and pelvic tilt angle was also collected.

### Pelvic tilt angle

To measure the pelvic tilt angle, we used a bubble inclinometer and a palpation meter (PALM; Performance Attainment Associations, St. Paul, MN, USA) equipped with two caliper arms. The bubble inclinometer, which has a semicircular arc ranging from 0° to 30° on either side of the midline, was utilized for the measurements. Participants stood with their feet 30 cm apart and were instructed to focus on a fixed point to maintain postural stability. In an upright position with weight evenly distributed and arms crossed over the chest, the investigator palpated the anterior superior iliac spine (ASIS) and posterior superior iliac spine (PSIS). The pelvic tilt angle in this standing position was defined as the angle formed by a horizontal line between the ASIS and PSIS. Positive values indicated an anterior pelvic tilt, while negative values signified a posterior pelvic tilt. To ensure accuracy, three measurements were taken on each side (dominant and non-dominant), and the average of these measurements was calculated. The PALM device demonstrated excellent intra-examiner reliability and good inter-examiner reliability (*Herrington, 2011*).

## Outcome measures

### Hip range of motion

Hip ROM was assessed for both passive and active movements using a smartphone inclinometer application (Goniometer App, iOS). For hip internal (IR0) and external rotation (ER0) in a neutral position, participants lay prone. In contrast, a sitting position was used to evaluate IR90 and ER90 at 9° hip flexion. During each measurement, the investigator stabilized the surrounding joints of the lumbopelvic region and the knee with one hand while passively maneuvering the lower limb to determine the final ROM of the

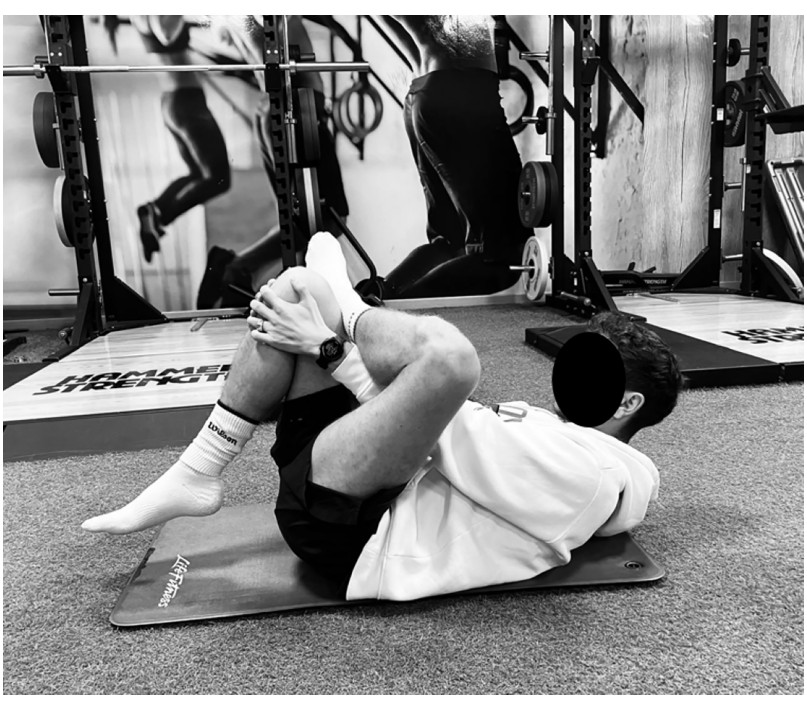

**Figure 3 Gluteal stretching group (SG).** 

hip, defined by a distinct, firm end feel without additional pelvic motion. For active hip ROM, participants moved their limb until they reached their maximum capability. The degrees for each measurement were meticulously recorded once the movement was completed. Participants were randomly assigned to start with either active or passive hip ROM. The smartphone was positioned along the midline of the tibia for all measurements. Each movement (both passive and active) was measured three times on both hips (left and right), and the mean value was calculated. The TRROM was calculated as the sum of internal and external rotation in both neutral and 90-degree hip flexion. Previous research has validated the reliability of using the inclinometer for assessing both passive and active hip ROM in prone and seated positions (*Han et al., 2015*; *Gradoz et al., 2018*).

### Active knee extension test

For the active knee extension (AKE) test, the participants assumed a supine position on a bench and flexed their knee and hip to 90 degrees. They closely monitored the position of the femur with their hand, strictly following the instruction not to permit any movement of the femur away from the hand throughout the test. The participant was directed to extend their leg as far as possible while keeping their foot relaxed, maintaining the extended position for 5 s. Each participant executed a preliminary repetition to familiarize themselves with the movement. A second repetition was then performed, and at the conclusion of the 5-s holding period, the angle of knee extension was measured using a standard goniometer (Physiomed, Manchester, UK). The center of the goniometer was aligned over the previously marked axis point on the lateral joint line, with the goniometer arms positioned along the lines marked on the femur and fibula. The goniometer

measurement was recorded within 2 s of reaching the end range of knee extension to ensure consistent duration of static stretch for each subject. AKE Test has shown an excellent reliability to measure hamstring flexibility (*Hamid, Mohamed Ali & Yusof, 2013*).

### Active straight leg raise

For the active straight leg raise (ASLR) test, a standard examination table was utilized. Participants lay supine and were instructed to remain relaxed throughout the test. The participants flexed the tested limb with the knee fully extended and the foot in a relaxed position, while the examiner secured the contralateral limb in a fully extended and neutrally rotated position. The movement ceased either when the tester encountered strong resistance or when pelvic rotation became apparent. The goniometer was positioned over the greater trochanter, with one arm aligned with the lateral femoral condyle and the other parallel to the table, pointing toward the mid-axillary line. ASLR showed excellent reliability to measure the posterior chain flexibility (*Neto et al., 2015*).

The pre- and post-measurements of the outcome measures were conducted randomly to each participant to minimize measurement bias and ensure the reliability of the results. This randomized approach helped maintain the integrity of the data collection process.

## Statistical analysis

IBM SPSS Statistics version 29.0 software for Windows (IBM, Armonk, NY, USA) was used for the statistical analysis. To evaluate data distribution, the Kolmogorov-Smirnov test and histogram examination were employed. Data were presented as mean and standard deviation. To compare the baseline characteristics of the two groups, an independent t-test was conducted and considering assumptions of homoskedasticity and sphericity. If these assumptions were met, a two-way analysis of variance (ANOVA) with a $2 \times 2$ design was carried out. The effect size was evaluated using partial eta squared ($\eta^2p$), with values of 0.01 interpreted as small, 0.06 as medium, and 0.14 as large. The percentage change was calculated with the following formula: ((baseline–post-intervention)/baseline) * 100 (*Hopkins et al., 2009*). A 95% confidence interval was set to all analyses.

## RESULTS

Of the 33 volunteers were screened, two of them were excluded ($n = 2$ did not meet the age criteria) (Fig. 4). Therefore, a total of 31 participants (62 lower limbs) were included in the study. Table 1 summarizes the anthropometric data of age, height, weight, BMI, femur length and pelvic tilt of the participants. No significant differences between groups were observed at baseline for all the dependent variables (active and passive ROM, ASLR and AKE).

## Hip external and internal rotation active and passive ROM

There was a significant difference in active ROM for the time effect in the ER0 ($p = 0.004$), IR0 ($p = 0.004$), IR90 ($p < 0.001$) and TRROM in neutral position ($p < 0.001$) but not in ER90 and TRROM at 90 degree of hip flexion ($p > 0.05$). Group-by-time interaction showed significant differences between groups in ER90 ($p = 0.001$) and TRROM ($p < 0.001$) at 90 degree of hip flexion, with a large effect size ($\eta^2p = 0.156$ and 0.197,
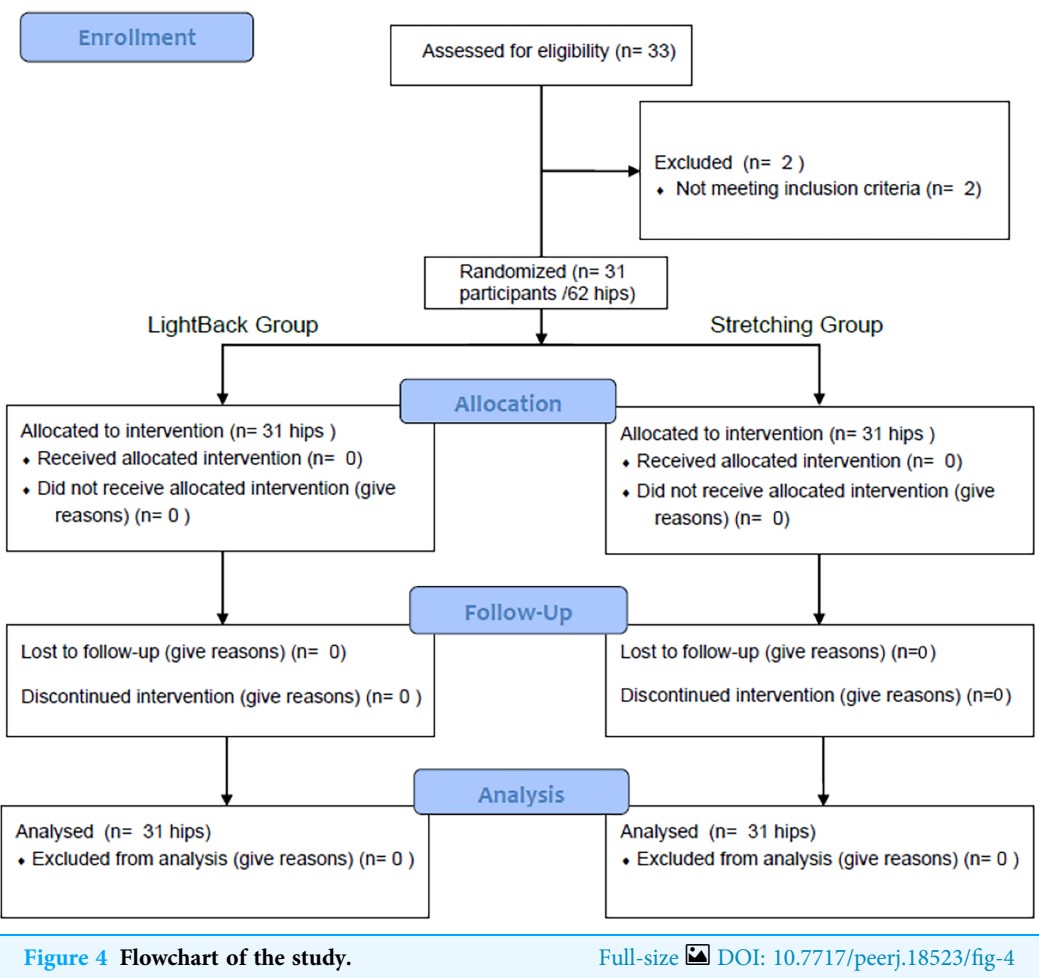

**Figure 4  Flowchart of the study.**

respectively), but not significant in the rest of active ROM ($p > 0.05$). *Post-hoc* analysis revealed that the LBG had large and significant active ROM in ER90 ($p < 0.001$), ER0 ($p = 0.002$), IR0 ($p = 0.007$), TRROM at neutral and 90 degrees of hip flexion ($p < 0.001$), when compared to the SG.

There was a significant difference in passive ROM for the time effect in all the variables ($p < 0.05$). Group-by-time interaction showed significant differences between groups with medium to large effect size in ER0 ($p = 0.034$; $\eta^2 p = 0.073$), IR90 ($p = 0.020$; $\eta^2 p = 0.088$), TRROM in neutral ($p = 0.019$; $\eta^2 p = 0.089$) and at 90 degrees of hip flexion ($p = 0.044$; $\eta^2 p = 0.066$). *Post-hoc* analysis showed that the LBG had significant and large improvement for ER90 ($p = 0.004$), ER0 ($p < 0.001$), IR90 ($p = 0.001$), IR0 ($p < 0.001$), TRROM at neutral and 90 degrees of hip flexion ($p < 0.001$), when compared to the SG.

## AKE and ASLR

Time effect showed significant differences in both groups for AKE and ASLR ($p < 0.001$). Group-by-time interaction showed significant differences between groups with a large effect size in ASLR ($p < 0.001$; $\eta^2 p = 0.161$) but not in AKE measurements. *Post-hoc* analysis revealed significant and large improvements in LBG for AKE and ASLR ($p < 0.001$) when compared to SG.

**Table 1 Descriptive data of the total sample.**

| Anthropometric data | Total sample ($n = 31$) |
| --- | --- |
| Age (years) | 20.42 ± 1.81 |
| Height (m) | 1.75 ± 0.11 |
| Weight (kg) | 69.79 ± 12.84 |
| BMI (kg/m$^2$) | 22.70 ± 2.68 |
| Femur length (cm) | 43.02 ± 2.86 |
| Pelvic tilt (degree) | 9.07 ± 3.60 |

Note:
Data are expressed as mean (SD).

Tables 2–4 show the mean values, percentage of change and the rest of statistics of both groups at baseline and immediate-post intervention.

## DISCUSSION

### Hip range of motion

While several studies have investigated the effects of static stretching on hip ROM (*Moreside & McGill, 2012*; *Moreside & Mcgill, 2013*; *Hammer et al., 2017*; *Hatefi, Babakhani & Ashrafizadeh, 2021*; *Yan, Jiang & Song, 2024*). This study demonstrated significant improvements in both active and passive hip ROM across various positions in the LBG, while the stretching group (SG) showed a reduction in active ROM in several cases, such as internal and external rotation ROM. These decreases in ROM in the SG group could be attributed to the effect of static stretching, which can temporarily reduce muscle strength and neuromuscular activation, leading to a short-term decline in active ROM (*Kay & Blazevich, 2012*). Static stretching is commonly used to relax muscles and soft tissues, enhance joint range of motion, and address stiffness, tightness, and rigidity (*Thomas et al., 2018*). In contrast, the Lightback system applies controlled axial forces, which appear to avoid this decline and may even enhance flexibility without compromising strength. This is a key advantage of the Lightback system, as it provides an improvement in both active and passive ROM without inducing the performance losses commonly associated with static stretching, but further research is needed to confirm our findings.

When comparing our results with those of *Yan, Jiang & Song (2024)* and *Moreside & McGill (2012)*, both studies reported more substantial improvements in hip ROM than observed in our study. *Yan, Jiang & Song*'s *(2024)* static stretching group demonstrated significant increases in internal rotation (24.4%) and external rotation (26.5%), while *Moreside & McGill (2012)* found even larger ROM gains of up to 56% in hip rotation. In our study, SG experienced minimal changes, with a slight decrease in active internal rotation (−1.56%) and a modest increase in active ER (2.94%). Meanwhile, LBG showed more consistent improvements with a 5.99% increase in IR and an 8.43% increase in ER, though these were still lower than those reported in the other studies. The key differences likely stem from the longer intervention durations in both studies (6 weeks *vs.* the acute effects). Furthermore, the nature of the interventions varied, with *Moreside & McGill (2012)* incorporating a combination of static and ballistic stretching

Table 2 Active ROM measures at baseline and immediately post intervention.

| Variables | Group | Baseline | Immediate-post | Percentage change (%) | $p$-value (time) | F; $p$-value (time x group) | $p$-value (post-hoc) | $\eta^2$p (time x group) |
|---|---|---|---|---|---|---|---|---|
| ER 90 | LBG | 33.74 (5.49) | 36.26 (5.46) | 7.47 | 0.058 | 11.074; $p = 0.001$ | <0.001 | 0.156 |
|  | SG | 34.13 (4.07) | 33.45 (5.00) | −1.99 |  |  | 0.322 |  |
| ER 0 | LBG | 43.52 (7.97) | 47.19 (9.76) | 8.43 | 0.004 | 2.187; $p = 0.144$ | 0.002 | 0.035 |
|  | SG | 42.77 (8.32) | 44.03 (9.09) | 2.94 |  |  | 0.281 |  |
| IR 90 | LBG | 38.23 (6.93) | 39.39 (6.80) | 3.03 | 0.004 | 2.634; $p = 0.110$ | 0.138 | 0.042 |
|  | SG | 39.13 (6.93) | 38.52 (5.93) | −1.56 |  |  | 0.431 |  |
| IR 0 | LBG | 40.94 (11.07) | 43.39 (11.53) | 5.99 | <0.001 | 0.056; $p = 0.815$ | 0.007 | 0.001 |
|  | SG | 40.77 (10.42) | 42.94 (10.40) | 5.34 |  |  | 0.016 |  |
| TRROM 90 | LBG | 71.97 (8.76) | 75.71 (7.45) | 5.20 | 0.077 | 14.740; $p < 0.001$ | <0.001 | 0.197 |
|  | SG | 73.26 (6.75) | 71.90 (6.25) | −1.86 |  |  | 0.154 |  |
| TRROM 0 | LBG | 84.61 (11.62) | 90.58 (12.86) | 7.07 | <0.001 | 1.870; $p = 0.177$ | <0.001 | 0.030 |
|  | SG | 83.87 (13.82) | 86.97 (14.27) | 3.70 |  |  | 0.041 |  |

Note:
Abbreviations: ER, external rotation; IR, internal rotation; ROM, range of motion; TRROM, total rotation range of motion. Data are expressed as mean (SD).

and *Yan, Jiang & Song (2024)* focusing on a hypomobility syndrome population. These methodological and participant differences likely contributed to the greater improvements seen in their studies compared to the more moderate gains in our acute interventions. This study is the first to evaluate the effects of the Lightback system and an isolated gluteal stretch on hip ROM. The findings demonstrate that both interventions are useful for improving ROM, particularly in passive ROM, where the greatest gains were observed. These results highlight the potential of both the Lightback system and gluteal stretching in providing acute improvements in hip flexibility, with the added benefit of avoiding the performance declines often associated with static stretching.

Therefore, based on the results of our study, the Lightback system's non-invasive approach to improving hip ROM presents a lower-risk alternative, particularly valuable for individuals where conventional femoroacetabular management strategies based on mobility work are inadequate or contraindicated (*González-de-la-Flor, 2024*). In this sense, short-amplitude static stretches, as in the case of the Lightback system, may be proposed with the aim of reducing possible damage to performance derived from static stretching in large hip ROM positions (*Hammer et al., 2017*).

## Posterior chain flexibility

In terms of posterior chain flexibility, our study measured both AKE and ASLR, which are commonly used to assess hamstring flexibility and overall posterior chain mobility. Several studies have demonstrated that proprioceptive neuromuscular facilitation (PNF) stretching improves hamstring flexibility and posterior chain mobility (*Kay & Blazevich, 2012*; *Cai, Liu & Li, 2023*). However, our study is one of the first to show significant improvements in both hip ROM and posterior chain flexibility using the Lightback system.

In our results, the LBG showed substantial improvements in both AKE and ASLR tests, with percentage increases of 24.54% and 23.19%, respectively. In comparison, while the SG

**Table 3 Passive ROM measures at baseline and immediately post intervention.**

| Variables | Group | Baseline | Immediate-post | Percentage change (%) | p-value (time) | F; p-value (time x group) | p-value (post-hoc) | $\eta^2$p (time x group) |
|-----------|-------|----------|----------------|----------------------|----------------|--------------------------|-------------------|--------------------------|
| ER 90 | **LBG** | 42.90 (5.84) | 45.39 (6.23) | 5.77 | <0.001 | 0.462; p = 0.500 | 0.004 | 0.008 |
| | **SG** | 40.77 (5.26) | 42.45 (6.68) | 4.12 | | | 0.050 | |
| ER 0 | **LBG** | 48.00 (9.53) | 53.55 (9.76) | 11.56 | <0.001 | 4.720; p = 0.034 | <0.001 | 0.073 |
| | **SG** | 47.45 (9.16) | 49.84 (9.26) | 5.04 | | | 0.024 | |
| IR 90 | **LBG** | 44.13 (8.17) | 46.61 (8.53) | 5.63 | 0.023 | 5.757; p = 0.020 | 0.001 | 0.088 |
| | **SG** | 45.19 (6.54) | 45.16 (7.12) | −0.07 | | | 0.965 | |
| IR 0 | **LBG** | 47.61 (11.52) | 50.68 (11.98) | 6.41 | <0.001 | 2.836; p = 0.097 | <0.001 | 0.045 |
| | **SG** | 47.06 (9.54) | 48.35 (9.85) | 2.74 | | | 0.088 | |
| TRROM 90 | **LBG** | 87.03 (8.97) | 91.90 (8.81) | 5.61 | <0.001 | 4.248; p = 0.044 | <0.001 | 0.066 |
| | **SG** | 85.97 (7.02) | 87.71 (7.16) | 2.02 | | | 0.110 | |
| TRROM 0 | **LBG** | 95.61 (13.66) | 104.23 (15.13) | 8.98 | <0.001 | 5.860; p = 0.019 | <0.001 | 0.089 |
| | **SG** | 94.52 (13.21) | 98.19 (13.40) | 3.89 | | | 0.013 | |

Note:
Abbreviations: ER, external rotation; IR, internal rotation; ROM, range of motion; TRROM, total rotation range of motion. Data are expressed as mean (SD).

**Table 4 AKE and ASLR measures at baseline and immediately post intervention.**

| Variables | Group | Baseline | Immediate-post | Percentage change (%) | p-value (time) | F; p-value (time x group) | p-value (post-hoc) | $\eta^2$p (time x group) |
|-----------|-------|----------|----------------|----------------------|----------------|--------------------------|-------------------|--------------------------|
| AKE | **LBG** | 24.42 (11.79) | 18.42 (10.41) | −24.54 | <0.001 | 3.892; p = 0.053 | <0.001 | 0.061 |
| | **SG** | 23.65 (9.53) | 20.71 (10.50) | −12.44 | | | 0.010 | |
| ASLR | **LBG** | 21.74 (11.85) | 16.68 (12.61) | −23.19 | <0.001 | 11.537; p = 0.001 | <0.001 | 0.161 |
| | **SG** | 21.48 (8.32) | 20.19 (12.13) | −6.00 | | | 0.106 | |

Note:
Abbreviations: AKE, active knee extension; ASLR, active straight leg raise. Data are expressed as mean (SD).

also showed improvements in AKE (−12.44%), the improvement in ASLR (−6.00%) was notably smaller and statistically insignificant. This discrepancy could be explained by the nature of the stretches. The AKE test predominantly focuses on hamstring flexibility, which responds well to static stretching interventions, while the ASLR test involves a more hip-dominant stretch that targets the posterior chain globally. The Lightback system likely had a more profound effect on the deep gluteal and pelvic structures, which are more engaged in the ASLR test, thereby leading to the significant improvements in the LBG group. Additionally, the distinction between the AKE and ASLR tests lies in the biomechanical focus of each movement. The AKE test is primarily knee-dominant, targeting the hamstring muscles more directly, while the ASLR is hip-dominant and incorporates the entire posterior chain, including the hip and lower back. This may explain why the Lightback system, which applies axial forces that mobilize the hip joint, was more effective in improving ASLR results. In contrast, the SG group, which employed traditional static stretching, showed more modest improvements in both tests, likely due to the limited engagement of the deep gluteal and pelvic structures.

When comparing our findings with previous studies (*Iwata et al., 2019*; *Villers et al., 2022*; *Cai, Liu & Li, 2023*; *Železnik et al., 2024*), the Lightback system had similar

improvements in posterior chain flexibility. *Villers et al. (2022)* reported a 4.6° increase in AKE after static stretching, which is lower than the 6° improvement seen in our LBG. This difference likely stems from the more comprehensive engagement of the posterior chain structures by the Lightback system, which targets not only the hamstrings but also the gluteal and pelvic regions. Similarly, *Iwata et al. (2019)* found a 10% increase in ROM following dynamic stretching, whereas our LBG achieved a comparable 23.19% improvement in ASLR. *Železnik et al. (2024)* reported improvements in posterior chain flexibility following both static stretching and proprioceptive neuromuscular facilitation stretching (hold-relax), with similar values as obtained in our study, showing that several stretching strategies are effective to improve posterior chain flexibility.

Interestingly, although our study used the Lightback system and a gluteal static stretch, the improvements in posterior chain flexibility, especially in ASLR, suggest that indirect engagement of the hamstrings and surrounding structures may have played a role (*Mendiguchia et al., 2020*). One possibility is that the Lightback system may have influenced neural dynamics, specifically through mechanisms related to the mechano-sensitivity of the sciatic nerve, which could have contributed to the enhanced flexibility across the posterior chain (*Kornberg & Lew, 1989*; *Balcı et al., 2020*). However, this hypothesis remains speculative and requires further investigation.

These findings are important, as hamstring flexibility is often a key factor in musculoskeletal health, with reduced flexibility being a common risk factor for conditions such as hamstring strains (*Malliaropoulos et al., 2012*; *Mendiguchia et al., 2021*). By improving both hip and knee ROM, the Lightback system may offer a more comprehensive approach to enhancing flexibility. Future research could explore these effects over a longer duration to assess whether these acute improvements translate into long-term benefits or adding lumbar joint mobilizations (*Villers et al., 2022*).

## Limitations

This study acknowledges several limitations that may influence the interpretation and generalizability of its findings. Firstly, the research was conducted on a homogeneous group of healthy, young adults from the European University of Madrid, which may limit the applicability of the results to older populations or those with existing musculoskeletal or health conditions. Participants' prior experience with stretching or similar exercises was not formally controlled or accounted for in the analysis, which may have influenced their response to the interventions. Although hip ROM measurements were conducted with high reliability, previous studies have reported error margins ranging from 0.9° to 3.3° under controlled conditions (*Fraeulin et al., 2020*). These potential variations in measurement accuracy could introduce slight inconsistencies in the reported ROM changes, though they are unlikely to significantly alter the overall outcomes of the study. Nevertheless, the error margin should be considered when interpreting the results, particularly in cases where the differences in ROM are small. Additionally, the intervention's efficacy was tested using the Lightback system, a specific device that might not be widely available or applicable in all clinical or rehabilitation settings, potentially limiting the broader applicability of these findings. Furthermore, the study's scope was

focused on specific outcomes related to hip ROM and posterior chain flexibility and did not account for long-term impacts or the sustainability of improvements post-intervention. Future research is called to explore these limitations, particularly the need for studies with diverse populations, varied intervention tools, and assessment of long-term outcomes to enhance the robustness and applicability of the findings.

## CONCLUSION

The findings of this study demonstrated that both the Lightback system and static gluteal stretching improved passive hip ROM. However, the Lightback system showed greater improvements in active ROM, particularly in external and internal rotation at various degrees of hip flexion, as well as in posterior chain flexibility. Notably, the Lightback system significantly enhanced large improvement in the ASLR test.

### Funding
The authors received no funding for this work.

### Competing Interests
The authors declare that they have no competing interests.

### Author Contributions
- Charles Cotteret conceived and designed the experiments, performed the experiments, authored or reviewed drafts of the article, and approved the final draft.
- Jaime Almazán-Polo conceived and designed the experiments, performed the experiments, authored or reviewed drafts of the article, and approved the final draft.
- Fabien Guérineau conceived and designed the experiments, performed the experiments, authored or reviewed drafts of the article, and approved the final draft.
- Ángel González de-la-Flor conceived and designed the experiments, analyzed the data, prepared figures and/or tables, authored or reviewed drafts of the article, and approved the final draft.

### Human Ethics
The following information was supplied relating to ethical approvals (*i.e.*, approving body and any reference numbers):
   The Universidad Europea de Madrid granted Ethical approval to carry out the study within its facilities (CIPI: 2023-296).

### Clinical Trial Ethics
The following information was supplied relating to ethical approvals (*i.e.*, approving body and any reference numbers):
   The Universidad Europea de Madrid granted Ethical approval to carry out the study within its facilities (CIPI: 2023-296)

## Data Availability

The raw data is available in the Supplemental File.

## Clinical Trial Registration

The following information was supplied regarding Clinical Trial registration:

ANZCTR: ACTRN12624000300572.

## Supplemental Information

Supplemental information for this article can be found online at http://dx.doi.org/10.7717/peerj.18523#supplemental-information.

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
