# Peer review of "The effects of gluteal stretching vs. Lightback® on hip rotation range of motion and posterior chain flexibility in healthy subjects: a cross-over clinical trial"

_PeerJ, doi:10.7717/peerj.18523_

## Round 0.1 · original submission · Major Revisions

I recommend that the authors explain in detail the methods used as commented by some reviewers.

Reviewer 1 ·

Basic reporting

In the abstract of the study, the problem and novelty were unclear.
The LightBack (LB) system was introduced suddenly, its purpose should be clearly stated.
It was also difficult to understand what was done in the methods section.
The methods should be described clearly.
Since I could not understand the methods, I could not proceed to read the results and conclusions.
The overall paper needs to be reorganized.

Experimental design

None.

Validity of the findings

None.

Additional comments

None.

Reviewer 2 ·

Basic reporting

no comment

Experimental design

#1. The reason for comparing LightBack stretching with conventional Gluteal stretching in your study is not clearly stated in the Introduction section. What are the specific issues with conventional Gluteal stretching? Has it been previously reported that LightBack stretching might address these issues? The knowledge gap that your study aims to fill is unclear, making it difficult to assess whether the chosen study design (cross-over RCT) is appropriate for your research objectives.

Validity of the findings

no comment

Additional comments

#2. Your study does not qualify as a superiority trial. It is unclear whether the results exceed a superiority margin.
#3. Lines 258-270 contain logical leaps. The conclusions drawn in this section cannot be supported by the results of your study.
#4. It is unclear from your study results whether the intervention is effective for the prevention or treatment of injuries. There is an overextension in the conclusions drawn.

Reviewer 3 ·

Basic reporting

The manuscript is well-structured and adheres to the journal's guidelines. Figures and tables are well-labeled, and the raw data is appropriately shared. However, there is a slight lack of clarity in the description of the LightBack system that might confuse readers unfamiliar with the device. English improvements are required.

Experimental design

The study design is robust, with a randomized crossover trial approach that is suitable for comparing the effects of gluteal stretching and the LightBack system. The methods are described in detail, allowing for potential replication. The inclusion and exclusion criteria are well-defined, and the sample size calculation is justified using a power analysis. The interventions are properly controlled, but the manuscript could benefit from a more thorough discussion of potential confounding factors, such as the participants' prior experience with stretching or similar exercises.

Validity of the findings

The findings are presented clearly and supported by statistical analysis. The study demonstrates significant differences between the interventions, with the LightBack system showing a greater impact on hip rotation range of motion (ROM) and posterior chain flexibility. The conclusions are consistent with the results, but the manuscript could explore the broader implications of these findings more thoroughly. The discussion section effectively relates the study results to existing literature, but it could benefit from a more critical analysis of the limitations, particularly regarding the short-term nature of the interventions and the homogeneity of the sample.

Additional comments

Abstract

The abstract lacks clarity by not defining "TRROM" and failing to specify which method, LB or SG, showed superior results in the significant differences mentioned. It should explicitly compare the effectiveness of LB and SG, as the title suggests, and reflect this comparison in the conclusion. The focus should be on providing a clear, concise comparison between the methods, ensuring the conclusion addresses both interventions rather than just LB. The abstract should be rewritten to ensure that it provides a clear and concise comparison between the two methods, which is the central theme of the study. This includes explicitly stating which method was more effective in each of the outcomes measured and reflecting this comparison in the conclusion. The revised abstract should also briefly touch on the practical implications of the findings, particularly for clinicians or practitioners who might choose between these two methods based on the study results.

Introduction

In line 32, you mention TROM as a risk factor for hip and groin injuries, but it's unclear whether this applies to TRROM, which is the focus of your study. You should evaluate and discuss the impact of TRROM on groin health and injury prevention, ensuring that your study's relevance is clear.
Lines 36 and 43 are repetitive, as are lines 55 and 57 where "limited hip rotation of the hip" is redundantly stated. These repetitions should be eliminated for clarity.
The hypothesis presented in lines 55-57 needs to be directly tied to your study's results. If your study does not address this hypothesis, it should be rephrased or omitted to maintain relevance.
Sample size calculation
The sample size was calculated independently using G*Power, which is a robust approach. However, referencing similar studies or clinical trials that have employed comparable devices or interventions could have provided additional context and validation for this calculation. Additionally, it's important to consider the participants' prior experience with stretching, as this could influence the outcomes and should be included as a factor in the study design or analysis.

Procedure

It's important to clarify whether the pelvic position during the Gluteal stretch was monitored, specifically if both ischial tuberosities were kept in contact with the ground. Monitoring this aspect is crucial because it ensures consistency in the stretch application, preventing any unintended variations in the stretch's effectiveness. If this was not controlled, it could introduce variability in the results, potentially impacting the study's conclusions. If it was monitored, this should be explicitly stated in the methods section to confirm the standardization of the stretching procedure.
In line 164, it's important to specify whether any backrest or other method was used to ensure that the 90-degree position was consistently maintained during the measurements. Maintaining this angle is critical for the accuracy of the hip rotation range of motion (ROM) assessments.

Results

It’s important to clarify whether the person measuring passive ROM was blinded to the intervention group. Lack of blinding could introduce bias, as personal preferences for a particular method might affect the results. This should be addressed to ensure the validity of the findings.
The experience level of the person measuring ROM should be specified. Highly experienced individuals reduce variability and increase the reliability of the measurements, which is crucial for the study's credibility.
The observed decrease in active ROM measures (TRROM 90, IR90, and ER90) after the Gluteal Stretching (SG) intervention is noteworthy. This phenomenon should be explored and explained further in the discussion to understand its implications fully.

Discussion

Discussion part is the weakest part of this article. The discussion lacks depth in relating the study's findings to the cited literature. While numerous studies are mentioned, the authors fail to connect these studies' results to their own findings meaningfully. The study should discuss the need to compare the LightBack method with dynamic stretching. The discussion should also include practical applications of the results.
Lines 242-246 mention studies indicating the importance of hip rotation ROM in preventing LBP and groin or hip-related pain. However, the authors did not measure LBP or other pain outcomes in their study.
Line 249-257: Lines 249-257 discuss studies related to hip joint manipulation and mobilization, but the connection to the current study is unclear. The authors should clarify how these findings relate to the use of the LightBack system and its impact on hip ROM.
Line 274: Can you elaborate what can be the reason why there were no differences in ASLR, but significant differences were observed in AKE test?
In line 285, the discussion cites a study on the effects of stretching on MVC without providing a clear definition of what is considered "prolonged" stretching. The authors should clarify this and ensure that the citation accurately reflects the study's findings, particularly regarding the lack of negative effects on MVC after 60 seconds of stretching. (Adam M. Hammer, Roger L. Hammer, Karen V. Lomond, Paul O'Connor, Acute changes of hip joint range of motion using selected clinical stretching procedures: A randomized crossover study)
Lines 288-292 repeat information from earlier in the discussion (lines 284-287). This redundancy should be removed to make the discussion more concise and focused.

---

## Round 0.2 · Minor Revisions

I think the comments of Reviewer 3 are of great concern and should be adressed.

Reviewer 3 ·

Basic reporting

The manuscript has been generally corrected. There are only a few minor errors, such as:
in lines 123-127 the quotes are missing
in line 175 the abbreviations AKE and ASLR are used for the first time, which should be in full length.

Experimental design

One uncertainty still remains for me, if the participants in the Gluteal Stretching group had to keep the hip flexed at 90 degrees, how did they get to the point of stretching? With what kind of movement?
In line 311, please write what app you used.
What was the order of measurements? Was the AKE always after the ASLR? Could this order have affected the ASLR results?

Validity of the findings

no comment

Additional comments

In discussion lines 461-461 you should write that this statement needs further investigation.

---

## Round 0.3 · accepted · Accept

I confirm that the authors have adressed all of the reviewers' comments. Although the other reviewers have not reviewed the revised versions of the manuscript, I attest that the authors have responded to all issues raised by these reviewers. The manuscript is ready for publication.

Reviewer 3 ·

Basic reporting

no comment

Experimental design

no comment

Validity of the findings

no comment

Additional comments

no comment